# Impact of screen time during COVID-19 on eating habits, physical activity, sleep, and depression symptoms: A cross-sectional study in Indian adolescents

Panchali Moitra ◉°*, Jagmeet Madan°

Department of Food, Nutrition & Dietetics, Sir Vithaldas Thackersey College of Home Science (Autonomous), SNDT Women's University, Santacruz West, Mumbai, Maharashtra, India

° These authors contributed equally to this work.
* panchali2511@yahoo.co.in, panchali.moitra@svt.edu.in

**Data Availability Statement:** All relevant data are available on Figshare (DOIs: 10.6084/m9.figshare.16934107 and 10.6084/m9.figshare.15077496).

## Abstract

### Objective

This study attempted to address the limited knowledge regarding the impact of screen time (ST) on lifestyle behaviors in Indian adolescents during the ongoing COVID-19 pandemic. The objectives were to 1) evaluate frequency and duration of using screens, and screen addiction behaviors in 10–15 years old adolescents in Mumbai during the COVID-19 pandemic and 2) examine the association of ST with lifestyle behaviors- eating habits, snacking patterns, physical activity (PA) levels, sleep quality and depression symptoms.

### Methods

An online survey was completed between January and March 2021. Eating habits, snacking patterns, time spent in different screen-based activities, and screen addiction behaviors were reported. The PA levels, sleep quality, and depression symptoms were evaluated using the Physical Activity Questionnaire for Children/Adolescents (PAQ C/A), Pittsburg Sleep Quality Index (PSQI), and Patient Health Questionnaire-2 (PHQ-2) respectively. Multiple linear regression analyses were performed to determine the impact of ST on lifestyle behaviors.

### Results

Adolescents (n = 1298, $M_{age}$ 13.2(1.1), 53.3% boys) reported the mean weekday and weekend ST as 442.3 (201.5) minutes/d and 379.9 (178.2) minutes/d respectively. Overall, 33.4% spent > 6hours/d for studying or doing homework, 65.4% used social networking sites for at least 2–3 hours/d, and 70.7% agreed that ST had taken up the majority of their leisure time. Only 12% engaged in moderate to vigorous PA (PAQ C/A scores <2). More than half (52.5%) reported PSQI >5 indicating poor sleep quality and 8.6% scored ≥ 3 on PHQ-2 to suggest a risk of depression. A higher ST was associated with lower PA and

**Funding:** The authors received no specific funding for this work.

**Competing interests:** The authors have declared that no competing interests exist.

increased sleep problems and a greater screen addiction was inversely associated with the eating habit, PA, and sleep-related variables.

## Conclusion

The study draws attention to a high prevalence of excess ST and its impact on eating habits, PA levels, and sleep quality in Indian adolescents during the COVID-19 pandemic. Targeted health promotion interventions that encourage judicious use of screens for education and entertainment and emphasize the adverse health effects of excess ST are required.

## Introduction

With the advent of the global COVID-19 pandemic and a need to mitigate the spread and transmission of the coronavirus, many countries, including India, had to issue stay-at-home advisories and impose lockdown restrictions and social distancing protocols [1,2]. The extended closures of schools prompted the educational institutions to adopt online teaching-learning models for students and the movement bans resulted in greater use of screens for entertainment and social interactions. While these measures have helped to a certain extent to maintain normalcy, they have also inadvertently increased the usage of varied screens, such as television, computers, video gaming, and mobile devices, in adolescents, much beyond the recommended duration of two hours per day [1–5]. The negative health consequences of excess screen time (ST), such as weight gain, behavioral problems, and sleep disturbances are well documented [6–9]. So, the increase in ST during the ongoing COVID-19 pandemic is likely to pose a greater risk of adverse health outcomes in adolescents, many of whom may have already been engaging in unhealthy lifestyle behaviors. This brings forth a need to investigate the prevalence of excess ST during the pandemic and evaluate its potential influences on lifestyle behaviors, such as eating habits, physical activity (PA) levels, and sleep patterns in adolescents.

Few studies that assessed the impact of the COVID-19 pandemic on adolescent lifestyle behaviors have suggested a complex, nonlinear, and context-driven effect. For instance, a recent study observed an increase in both ST and habitual PA in adolescents during the pandemic [4], but another cross-sectional study reported a drastic reduction in PA and a substantial increase in ST [10]. Besides the influence of ST on PA levels, there is evidence that adolescents who spend more time using screens tend to have a lower intake of fruits and vegetables and higher consumption of energy-dense snacks [6,11–13]. However, a majority of these studies were conducted before the pandemic and little is known about the association between excessive ST and adolescents' eating habits during the COVID-19.

Excess ST has also been associated with poor sleep quality, daytime dysfunction, and an increased risk of anxiety and depression in adolescents [7,8,14]. While the evidence regarding the association of screen-based sedentary behaviors with sleep latency, and sleep insufficiency is unequivocal [6,15,16], the influence of ST on mental health seems to vary based on the duration and content of screen exposure in adolescents [17–19]. Given that unhealthy lifestyle behaviors, such as physical inactivity, unhealthy diet, and inadequate sleep tend to co-occur [20], the impact of excessive ST may transude into multiple behaviors of adolescents, thereby escalating the health risks during the pandemic.

To date, there is no clarity on how long the lockdown restrictions will continue or the COVID-19 pandemic will last. As the lockdown restrictions continue in India and the chances of a third wave to commence persist, the impact of the increased ST during the COVID-19 on

the lifestyle habits of adolescents must be better understood. This information can help customize ST recommendations and guide effective policies and interventions aimed at moderating the health risks associated with excess ST in adolescents. Yet studies investigating the prevalence and magnitude of ST during the ongoing pandemic and its association with lifestyle behaviors and mental health in adolescents in India are still lacking.

To the best of our knowledge, this study is the first to provide a comprehensive investigation into several lifestyle behaviors, such as eating habits, PA levels, sleep quality, and depressive symptoms among adolescents in India during the pandemic and to assess these behaviors as a function of ST. The primary aim was to assess the impact of ST during COVID-19 on lifestyle behaviors in Indian adolescents and the specific objectives were to 1) evaluate frequency and duration of using screens, and screen addiction behaviors in 10–15 years old adolescents in Mumbai during the COVID-19 pandemic and 2) examine the association of ST with eating habits, physical activity (PA) levels, sleep quality and depression symptoms.

## Methods

### Study design, setting, and adolescents

This cross-sectional study was conducted among 10–15 years old adolescents attending grades 6 to 10 of six private schools and four government-aided schools in the city of Mumbai, India. The study sites were selected using a purposive sampling method. An online survey was conducted to collect data as an in-person survey was not feasible due to the ongoing pandemic-induced closures of educational institutes in India since late March 2020 [2]. Information leaflets containing details of the study and a link to provide parental consent were sent to each of the participating schools and colleges. The parents were informed to provide consent within a week of receiving the information sheets. Adolescents who provided signed parental consent (n = 1512) were invited to join virtual meetings scheduled separately for each of the study sites. The online survey was completed by 1298 adolescents in the presence of the investigators, research staff, and school representatives. Data were collected from January 2021 to March 2021 after obtaining ethical approval from an independent ethics committee, Intersystem Biomedica Committee, Mumbai (ISBEC protocol Version 1b, dated 16 December 2020).

### Sample size estimation

Based on a recent study that reported the prevalence of excessive ST (using screens >2hours/day) in urban adolescents in India as 68% [21] and after using 95% confidence level, a 5% margin of error, a non-response rate of 25%, and a proportional representation of adolescents from private and government schools, the final sample size was estimated as 805.

### Measures

An online survey including questions related to socio-demographic characteristics, eating habits, snacking behaviors, physical activity levels, screen time and screen addiction, sleep patterns, and depression symptoms were administered through google forms on a virtual meeting platform.

**Demographic characteristics.** Adolescents were asked to provide demographic information, such as gender, date of birth, class of study, type of living arrangement, father's present occupation, mother's working status, and type and number of screens (television/mobile phones/computers/laptops/tablets) owned by them and their families.

**Screen time.** In this study, the term 'screen time' indicates the time spent working/studying/playing using any screen device, and 'screen usage' refers to different screen devices, such

as laptops, mobile phones, television, and more, that were used by the adolescents. The type, frequency, and duration of screen usage were reported on a brief five-item questionnaire, that was developed by the researchers after an extensive review of similar instruments [7,17,22] used in previous studies among adolescents. The questionnaire administered to the adolescents is provided as Supplemental Material (File S1). In summary, the frequency of using different screens was reported from 'never (0 days)' to 'every day (7 days)', and the time spent using these screens on a typical weekday and a typical weekend was reported as minutes/d. To estimate the daily time spent in screen-related activities, the adolescents were asked '*In the last 7 days, how much time did you spend in the following screen-related activities*? and to evaluate the adolescents' addiction to screen usage, a five-point Likert scale (*strongly disagree to strongly agree*) was used. Additionally, two statements evaluated adolescents' perceived increase/decrease in their screen usage and ST during the lockdown.

**Eating habits and snacking patterns.**   To evaluate the eating habits, the adolescents were asked to report the frequency of consuming breakfast, having lunch or dinner with family, watching television while having meals, eating out with family and/or friends, and ordering takeaways in the past week. The response options—never, 1–2 days, 3–4 days, 5–6 days, and every day were scored 0–4. A brief 24 item food frequency questionnaire, that was validated in our previous study for the same population [23], estimated the consumption of fruits, freshly prepared fruit juices, packaged 100% fruit juices, vegetables (green leafy vegetables, orange and yellow vegetables, salad, and other vegetables), unhealthy snacks (foods high in fat, salt, and sugar) and carbonated beverages. Adolescents were asked '*In the last 7 days, how many days did you consume the following foods/beverages*?'. The responses were evaluated on a five-point scale, from never to 2 or more than twice a day, scored 0 to 4 for fruits and vegetables, and reverse coded as 4–0 for unhealthy snacks and carbonated beverages to ensure that higher scores indicated healthier eating habits. For each of the listed food items, the adolescents reported if their consumption has increased, decreased, or remained the same during the pandemic as compared to the pre-pandemic times.

**Physical activity levels.**   The validated self-reported instruments, the Physical Activity Questionnaire for Children and/or Adolescents (PAQ-C/-A) assessed the physical activity levels of adolescents. The PAQ -C/A has been extensively used to evaluate general physical activity levels of children and adolescents in previous studies [24,25]. The PAQ C is typically administered in children, ages 8 to 14 years and comprises of 9 items providing a 7-day recall of the type and frequency of activities performed in spare time, during physical education (PE) classes and recess breaks, right after school, in the evenings, and on weekends. The PAQ-A for adolescents > 14 years is a modified version of PAQ-C that includes the same items except for the question regarding activities performed during recess. In both the questionnaires, each item is scored from 1 to 5 to derive item-specific composite activity scores. The mean of these composite scores is used to determine the PAQ summary score that ranges from 1–5 with higher scores ($\geq 2$) indicating moderate to vigorous level of PA and scores < 2 as light PA [25].

To determine the changes in the frequency and duration of screen usage, the frequency of intake of specific food items, and engagement in different physical activities during the COVID-19 pandemic as compared to before pandemic, the adolescents were asked to report whether they perceived the changes as increased, decreased or remained similar. The responses to these questions generated quantitative data that helped estimate the impact of the pandemic on the selected measures.

**Sleep quality.**   The sleep patterns including subjective sleep quality, sleep latency, sleep duration, habitual sleep efficiency, sleep disturbances, and daytime dysfunction were evaluated using the Pittsburg Sleep Quality Index (PSQI) [26]. The validity and reliability of PSQI to

identify sleep-related problems in adolescents have been established in previous studies [26–28]. The PSQI comprises Likert-type and open-ended questions that are scored from 0 to 3. Of a maximum score of 21, a total score > 5 is considered indicative of poor sleep quality.

**Depression symptoms.**  The frequency of experiencing depression symptoms was assessed using Patient Health Questionnaire-2 or PHQ-2, a widely accepted and validated screening tool for major depressive disorders among adolescents [29,30]. The PHQ-2 includes the two items that inquire about the frequency of 'having little pleasure in doing things' and 'feeling down, depressed or hopeless in the past two weeks. The frequency options for each item include 'not at all' to 'nearly every day' (scored 1 to 3). Adolescents reporting an overall score ≥ 3 are considered to be at risk for depression.

## Statistical analysis

All analyses were performed using SPSS version 24. Descriptive statistics were calculated as mean and standard deviations or n (%). Comparison of categorical and continuous variables was performed using the chi-squared tests and one-way ANOVA, stratified by sex and age categories (10–12 years and 13–15 years). Multiple linear regression analyses were performed to assess the association of ST items, such as frequency and duration of screen usage, screen-related activities, and screen addiction scores with the dependent variables—eating habits (total healthy eating habit score was calculated by aggregating eating habit and snacking behavior item scores), physical activity levels (mean summary PAQ-C/A scores), sleep patterns (mean global PSQI scores) and depression symptoms (mean PHQ 2 scores) while controlling for adolescents' age, sex, and type of school attended (used as a proxy for socioeconomic status) as covariates. Univariable regression analyses were first performed to determine the unadjusted effect of factors associated with each of the dependent variables. Next, the independent variables with a significance level <0.1 were entered into the mixed-effects multivariable regression models to determine the ST variables that were associated significantly with eating habits, PA, sleep, and depression symptoms. Analysis of residuals confirmed the assumptions of linearity and as indicated in previous studies, the lowest values of Akaike's and Schwarz's Bayesian information criteria measures were used to test the goodness of fit of the final models. To test multicollinearity, Pearson correlation coefficient (r) values > 0.5 or the variance inflation factor (VIF) value > 10 were used as the diagnostic tests. The VIF values ranged from 2.12 to 8.15 (mean 5.44) for the majority of variables, except for an ST-related activity item (reading/listening to music) and an ST addiction statement (*I use screens for a longer duration than is good for me*), so these variables were excluded in the final model. Results were reported as standardized regression coefficients (β) and standard error (SE). All tests were two-tailed and considered statistically significant at a p-value ≤ 0.05.

## Results

### Sample characteristics

Of 1512 adolescents for whom parental consent was provided, 89 were not present on the survey day and 155 had provided > 20% incomplete/implausible data in the online survey. So, the analyses were performed on the final sample (n = 1298, 85.8% of those with parental consent) of adolescents, aged 10–12 years (n = 724) and 13–15 years (n = 574). The mean age of the adolescents was 13.2 (1.2) years). Overall, 53.3% were boys, 60.2% attended private schools, and 78.9% and 70.9% mentioned their living arrangement as a nuclear family and the mother's working status as a homemaker respectively (Table 1). Almost all adolescents reported their families having a television (95.1%) and mobile/smartphones (93.4%). The most common types of screen devices owned by the adolescents were smart/mobile phones (68.5%), laptops (43.5%), and Tablets or iPads (12.5%).

**Table 1. Demographic characteristics of 10–15 years old adolescents in the study (n%).**

| Variables | Overall (n = 1298) |
|---|---|
| Gender | |
| Boys | 692 (53.3) |
| Girls | 606 (46.7) |
| Age categories | |
| 10–12 years | 724 (55.8) |
| 13–15 years | 574 (44.2) |
| Type of school attended | |
| Private school | 782 (60.2) |
| Government-aided school | 516 (39.8) |
| Type of living arrangement | |
| Single parent family | 19 (1.5) |
| Nuclear family | 1024 (78.9) |
| Joint family | 179 (13.8) |
| Extended family | 76 (5.9) |
| Father's present occupation | |
| Service | 459 (35.4) |
| Business | 389 (30.0) |
| Menial jobs | 234 (18.0) |
| Self-employed | 140 (10.8) |
| Does not know | 76 (5.9) |
| Mother's working status | |
| Works full time ($>$ 6h/day) | 256 (19.7) |
| Works part-time ($<$ 5 h/day) | 106 (8.2) |
| Homemaker | 920 (70.9) |
| Does not know | 16 (1.2) |
| Number of screen devices owned by the family (including television, mobile phone) | |
| $\leq 2$ | 12 (0.9) |
| 3–5 | 870 (67.0) |
| $\geq 6$ | 416 (32.0) |
| Type of screens owned by the family (Response -Yes) | |
| Television | 1235 (95.1) |
| Desktop computer | 321 (24.7) |
| Laptop | 689 (53.1) |
| Smartphone/Mobile phone | 1212 (93.4) |
| Tablets/iPads | 378 (29.1) |
| Others (X Box/PlayStation/smartwatch) | 181 (13.9) |
| Type of screens owned by the participant (Response -Yes) | |
| Television (in own bedroom) | 112 (8.6) |
| Desktop computer | 67 (5.2) |
| Laptop | 564 (43.5) |
| Smartphone/Mobile phone | 889 (68.5) |
| Tablets/iPads | 162 (12.5) |
| Others (game consoles/handheld video games) | 104 (8.0) |

## Screen time

Among adolescents, 37.7%, 30.2%, and 64.9% reported using television, laptops/desktop computers, and mobile phones every day in the past week for study or entertainment respectively.

The total time spent using screens on weekdays was 442.29 (201.5) minutes/d and during weekends was 379.90 (178.2) minutes/d. One third (33.4%) had spent > 6hours/d using screens for studying or doing homework, two thirds (65.4%) reported being on social networking sites 2–3 hours/d and a majority reported that their screen usage (85.6%) and ST (94.9%) had increased during the pandemic. For screen addiction behaviors, 70.7% agreed/strongly agreed that screen time takes up the majority of their leisure time and 20.5% reported that they prefer socializing online than meeting people face to face (Table 2).

## Eating habits, physical activity levels, sleep quality, and depression symptoms of adolescents

The mean frequency of breakfast consumption and having meals sitting together with family was reported as 2.46 (1.31) d/week and 2.55 (1.22) d/week respectively. Adolescents mentioned watching television/any other screen while eating 4–5 times/week and consuming fast foods, carbonated beverages, and foods high in fat content > 3times/week. Overall, the fresh fruit items were consumed 4–5 times in the last week but the consumption of salads and healthy snacks were only 1–2 times/week. Increased frequency of consumption of fast food (66.8%), foods high in sugar (56.5%), fried foods (48,9%), carbonated beverages (59.1%), and fruits (72.3%) during the pandemic were reported. Adolescents reported the frequency of having online physical education (PE) classes as a part of the school curriculum and being active during these PE classes to be < 2times/week. Almost all adolescents (88.6%) mentioned a decrease in PA during the pandemic and a majority (88%) had light PA levels (as assessed using PAQ C/A scores < 2). Overall, 52.5% and 8.6% had PSQI >5 and PHQ-2 scores ≥ 3 respectively. In Fig 1, we have provided the results of the changes in the frequency of screen usage, eating

**Table 2. Screen related behaviors of 10–15 years old adolescents (n = 1298) in the study.**

| Frequency of screen usage/week n (%) | | | | |
|---|---|---|---|---|
| **Variables** | Never (0 days) | Sometimes (1–2 days) | Often (3–4 days) | Frequently (5–6 days) | Every day (7 days) |
| Television | 103 (7.9) | 112 (8.6) | 246 (19.0) | 348 (26.8) | 489 (37.7) |
| Laptop/desktop computer | 301 (23.2) | 126 (9.7) | 212 (16.3) | 267 (20.6) | 392 (30.2) |
| Mobile/Smart phone | 38 (2.9) | 45 (3.5) | 59 (4.5) | 313 (24.1) | 843 (64.9) |
| Tablets/iPads | 712 (54.9) | 174 (13.4) | 152 (11.7) | 102 (7.9) | 158 (12.2) |
| **Time spent in screen-related activities/day n (%)** | | | | |
| **Variables** | <30 minutes/d | 30 minutes- 1hour | 1hour–2hour | > 2 hours | > 4 hours |
| Studying/doing homework | 6 (0.5) | 25 (1.9) | 462 (35.6) | 372 (28.7) | 433 (33.4) |
| Using social networking sites | 137 (10.6) | 312 (24.0) | 754 (58.1) | 62 (4.8) | 33 (2.5) |
| Playing games | 93 (7.2) | 192 (14.8) | 877 (67.6) | 98 (7.6) | 38 (2.9) |
| Watching movies/YouTube | 26 (2.0) | 58 (4.5) | 1051 (81.0) | 101 (7.8) | 62 (4.8) |
| Reading/listening to music | 588 (45.3) | 290 (22.3) | 362 (27.9) | 38 (2.9) | 20 (1.5) |
| **Screen addiction behaviors n (%)** | | | | |
| **Variables** | Strongly disagree | Disagree | Neither agree nor disagree | Agree | Strongly agree |
| I can't imagine going anywhere without my mobile devices | 172 (13.3) | 246 (19.0) | 371 (28.6) | 383 (29.5) | 126 (9.7) |
| Screen time takes up a majority of my leisure time | 62 (4.8) | 170 (13.1) | 148 (11.4) | 631 (48.6) | 287 (22.1) |
| I use screens for a longer duration than is good for me | 117 (9.0) | 297 (22.9) | 333 (25.7) | 347 (26.7) | 204 (15.7) |
| I prefer socializing online than meeting people face to face | 520 (40.1) | 389 (30.0) | 123 (9.5) | 74 (5.7) | 192 (14.8) |

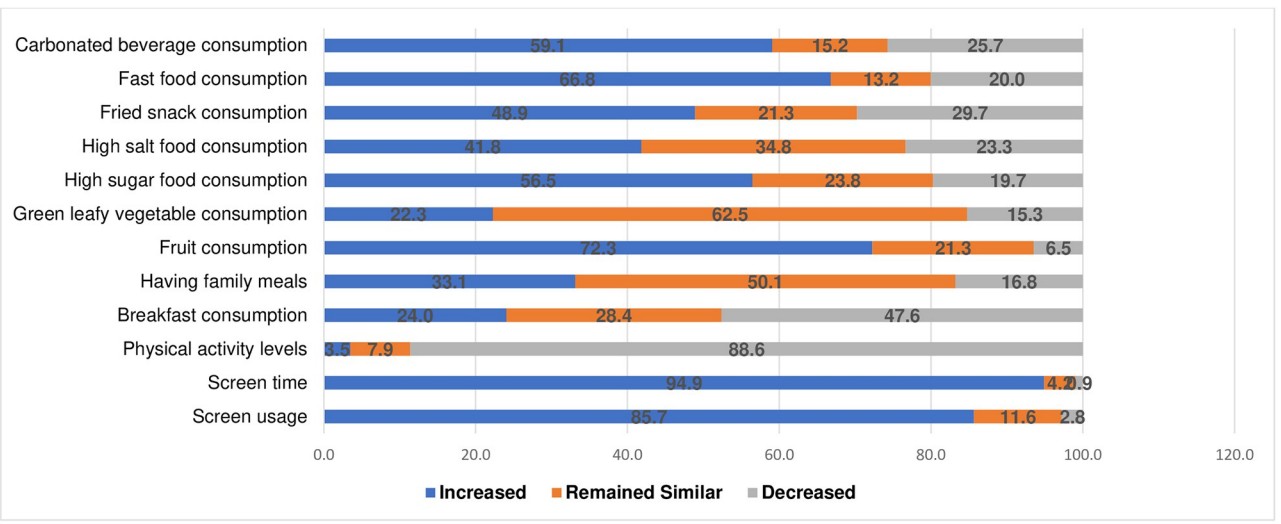

**Fig 1. Reported frequency of screen usage, eating habits, and physical activity levels before and during COVID-19.**

habits, snacking patterns, and physical activity levels during the pandemic as reported by the adolescents.

## Comparison between sex and age categories

No significant sex and age differences were observed in the eating habits and snacking patterns of adolescents except for the healthy snack consumption scores between girls (2.15 (0.89) d/week) and boys (1.76 (0.65) d/week, p = 0.033) (Table 3). For PA, we observed differences in the frequency of being active during PE classes and the mean PAQ C/A scores between boys and girls and between older and younger adolescents. Girls reported a higher frequency of using mobile/smartphones and using social networking sites as compared to boys. The mean scores for all screen addiction behavior-related statements were higher among older adolescents (13–15 years) as compared to the younger adolescents (10–12 years old). The mean sleep latency, daytime dysfunction, and global PSQI scores were higher among girls, and the mean depression symptom score for '*having little pleasure in doing things*' was higher among 13–15 years old adolescents. Also, the proportion of adolescents having poor sleep quality and depression symptoms increased with age (p < 0.05).

## Association of ST on lifestyle behaviors

The eating habit scores were observed to be associated with the time spent using social networking sites (β = -0.229 (0.110), p = 0.044) and the screen addiction score for '*ST takes a majority of my leisure time*' (β = -0.658 (0.222), p = 0.017) (Table 4). Increase in the frequency of screen usage (laptops/computers) was associated with a decrease in the eating habit (β = -0.298 (0.089), p = 0.034) and PA (β = -0.845 (0.212), p <0.001) scores and also with an increase in PSQI scores (β = 0.612 (0.334), p = 0.044), indicating poor sleep quality in the sample. The total time spent using screens on both weekdays (β = -0.343 (0.130), p = 0.006) and weekends (β = -0.831(0.119), p = 0.002) showed inverse associations with time spent in PA/d. Also, the analyses indicated that higher screen addiction behavior scores ('*I can't imagine going anywhere without my mobile devices*') were associated with reduced PA levels (β = -0.512 (0.215), p = 0.043) and poor sleep quality (β = 0.298 (0.089), p = 0.005) in adolescents.

**Table 3. Descriptive results of eating habits, screen time, physical activity levels and sleep quality during COVID-19 pandemic of 10–15 years old adolescents, stratified by sex and age categories.**

| Variables | Total (n = 1298) | Boys (n = 692) | Girls (n = 606) | P value | 10–12 years (n = 724) | 13–15 years (n = 574) | P value |
|---|---|---|---|---|---|---|---|
| **Eating habits (days/week)** [a] | | | | | | | |
| 1. Had breakfast | 2.46 (1.31) | 2.49 (1.29) | 2.39 (1.36) | 0.722 | 2.41 (1.47) | 2.52 (1.11) | 0.672 |
| 2. Had lunch/dinner sitting together with family | 2.55 (1.22) | 2.52 (1.20) | 2.61 (1.24) | 0.743 | 2.68 (1.22) | 2.40 (1.21) | 0.241 |
| 3. Had an evening meal in front of the Television/any screen | 4.76 (1.04) | 4.65 (1.02) | 6.00 (1.06) | 0.238 | 4.86 (1.04) | 4.65 (1.03) | 0.449 |
| 4. Brought fast foods from outside/Ordered takeaways | 0.58 (0.13) | 0.54 (0.19) | 0.70 (0.10) | 0.227 | 0.58 (0.12) | 0.59 (0.23) | 0.962 |
| **Snacking pattern (days/week)** [b] | | | | | | | |
| 1. Fresh Fruits | 4.71 (2.17) | 4.54 (2.66) | 5.06 (1.89) | 0.349 | 4.57 (2.21) | 4.87 (2.74) | 0.552 |
| 2. Salad | 1.50 (1.01) | 1.46 (1.01) | 1.58 (1.02) | 0.603 | 1.45 (0.97) | 1.56 (0.78) | 0.560 |
| 3. Healthy snacks | 1.94 (0.79) | 1.76 (0.65) | 2.15 (0.89) | **0.033*** | 1.89 (0.72) | 1.97 (0.81) | 0.221 |
| 4. High sugar foods | 2.94 (0.65) | 3.00 (0.65) | 2.82 (0.66) | 0.186 | 2.96 (0.62) | 2.92 (0.72) | 0.712 |
| 5. High-fat foods | 3.27 (0.60) | 3.23 (0.59) | 3.36 (0.60) | 0.272 | 3.29 (0.59) | 3.25 (0.60) | 0.762 |
| 6. High salt foods | 2.99 (0.71) | 3.03 (0.69) | 2.91 (0.72) | 0.424 | 2.96 (0.66) | 3.02 (0.75) | 0.685 |
| 7. Fast foods | 3.61 (0.56) | 3.62 (0.57) | 3.58 (0.56) | 0.714 | 3.64 (0.55) | 3.56 (0.58) | 0.472 |
| 8. Carbonated beverages | 3.68 (0.57) | 3.66 (0.55) | 3.73 (0.57) | 0.572 | 3.70 (0.44) | 3.63 (0.64) | 0.319 |
| **Physical activity levels** | | | | | | | |
| Frequency of online PE classes/week | 1.70 (0.92) | 1.85 (0.85) | 1.39 (0.99) | **0.037*** | 2.02 (0.62) | 1.33 (0.82) | **0.048*** |
| Frequency of being active in PE/week | 1.44 (1.41) | 1.63 (1.26) | 1.03 (1.63) | **0.020*** | 1.91 (1.05) | 0.90 (1.10) | **<0.001**** |
| Mean PAQ- C/A score | 1.15 (1.47) | 1.44 (1.05) | 0.85 (0.61) | **<0.001**** | 1.18 (0.71) | 1.10 (0.73) | **0.045*** |
| PA level (n%) | | | | | | | |
| Light PA (score <2) | 1142 (88.0) | 590 (85.3) | 552 (91.1) | **0.001*** | 629 (86.9) | 513 (89.4) | 0.168 |
| Moderate to vigorous (score >2) | 156 (12.0) | 102 (14.7) | 54 (8.9) | - | 95 (13.1) | 61 (10.6) | - |
| **Screen related behaviors** | | | | | | | |
| Frequency of screen usage (d/week) | | | | | | | |
| • Laptop/Computer | 5.12 (3.29) | 5.02 (2.89) | 5.22 (3.45) | 0.584 | 4.90 (2.28) | 5.46 (3.40) | 0.865 |
| • Mobile Phone | 6.67 (4.21) | 6.62 (3.99) | 6.69 (4.33) | 0.498 | 5.82 (3.89) | 7.02 (4.51) | **<0.001**** |
| • Television | 4.78 (3.64) | 4.99 (3.61) | 4.56 (2.89) | 0.123 | 5.12 (3.78) | 4.32 (3.23) | **<0.001**** |
| • Tablet/iPad | 1.23 (1.16) | 0.99 (0.88) | 1.69 (1.20) | **0.003*** | 1.22 (1.08) | 1.24 (1.20) | 0.539 |
| • Game consoles | 0.58 (0.12) | 0.61 (0.14) | 0.58 (0.09) | 0.821 | 0.62 (0.10) | 0.57 (0.15) | 0.291 |
| Time spent on screen-based activities [c] | | | | | | | |
| • Studying/doing homework | 2.57 (1.32) | 2.52 (1.36) | 2.67 (1.27) | 0.605 | 2.64 (1.30) | 2.52 (1.34) | 0.684 |
| • Using social networking sites | 1.57 (0.93) | 1.01 (0.87) | 2.00 (1.01) | **0.036*** | 1.05 (0.87) | 2.00 (0.99) | **<0.001**** |
| • Playing games | 1.93 (0.88) | 1.98 (0.82) | 1.78 (0.92) | 0.099 | 2.01 (0.85) | 1.88 (0.90) | 0.158 |
| • Watching movies/YouTube | 2.24 (1.16) | 2.17 (1.12) | 2.39 (1.16) | 0.219 | 2.21 (1.15) | 2.25 (1.16) | 0.312 |
| • Reading/listening to music | 0.56 (0.22) | 0.38 (0.20) | 0.72 (0.21) | **< 0.001**** | 0.48 (0.20) | 0.64 (0.24) | **0.002*** |
| Total screen time/weekday (minutes) | 442.29 (201.5) | 438.34 (178.2) | 451.12 (214.4) | 0.645 | 448.45 (167.34) | 431.11 (221.12) | 0.108 |
| Total screen time/weekend (minutes) | 379.90 (178.2) | 398.22 (166.8) | 365.30 (181.1) | 0.612 | 369.98 (166.23) | 385.20 (184.10) | 0.118 |
| **Screen addiction behaviors** [d] | | | | | | | |
| • I can't imagine going anywhere without my mobile devices | 2.01 (1.18) | 1.87 (1.25) | 2.30 (0.95) | 0.083 | 1.73 (1.18) | 2.33 (1.19) | **0.009*** |
| • Screen time takes up a majority of my leisure time | 2.68 (1.10) | 2.65 (1.20) | 2.76 (0.93) | 0.638 | 2.52 (1.08) | 2.88 (1.12) | 0.099 |
| • I use screens for a longer duration than is good for me | 2.17 (1.20) | 2.14 (1.28) | 2.24 (1.03) | 0.690 | 2.05 (1.22) | 2.31 (1.17) | 0.276 |

(*Continued*)

**Table 3.** (Continued)

| Variables | Total (n = 1298) | Boys (n = 692) | Girls (n = 606) | P value | 10–12 years (n = 724) | 13–15 years (n = 574) | P value |
|---|---|---|---|---|---|---|---|
| • I prefer socializing online | 1.21 (1.13) | 1.15 (1.04) | 1.33 (1.16) | 0.543 | 1.20 (1.10) | 1.21 (1.12) | 0.983 |
| **Sleep pattern using PSQI** | | | | | | | |
| Subjective sleep quality | 0.84 (0.53) | 0.80 (0.50) | 0.86 (0.56) | 0.218 | 0.77 (0.56) | 0.87 (0.46) | **0.044*** |
| Sleep Latency | 1.10 (1.05) | 0.87 (0.55) | 1.58 (1.09) | **< 0.001**** | 1.05 (0.98) | 1.12 (1.06) | 0.714 |
| Sleep Duration | 0.97 (0.68) | 0.98 (0.66) | 0.97 (0.69) | 0.443 | 0.85 (0.66) | 1.10 (0.71) | 0.836 |
| Sleep Efficiency | 0.89 (0.12) | 0.91 (0.11) | 0.88 (0.13) | 0.631 | 0.88 (0.10) | 0.89 (0.12) | 0.479 |
| Sleep Disturbances | 0.77 (0.28) | 0.82 (0.21) | 0.73 (0.29) | 0.335 | 0.82 (0.25) | 0.79 (0.29) | 0.492 |
| Daytime Dysfunction | 0.88 (0.41) | 0.65 (0.30) | 1.12 (0.55) | **0.027*** | 0.85 (0.40) | 0.92 (0.42) | 0.192 |
| Global PSQI score | 5.45 (3.21) | 5.21 (2.97) | 5.78 (3.30) | **0.011*** | 5.03 (3.02) | 5.82 (3.10) | **0.036*** |
| Mean Global PSQI score > 5 (n%) | 682 (52.5) | 340 (49.1) | 342 (56.4) | **0.008*** | 360 (49.7) | 322 (56.1) | **0.021*** |
| **Depression symptoms using PHQ-2** | | | | | | | |
| Little pleasure in doing things | 1.24 (0.65) | 1.27 (0.59) | 1.23 (0.72) | 0.992 | 0.89 (0.59) | 1.58 (0.68) | **<0.001**** |
| Feeling down, depressed or hopeless | 0.64 (0.33) | 0.63 (0.30) | 0.64 (0.34) | 0.712 | 0.62 (0.28) | 0.65 (0.35) | 0.760 |
| Mean PHQ-2 score ≥ 3 (n%) | 111 (8.6) | 58 (8.4) | 53 (8.7) | 0.847 | 43 (5.9) | 68 (11.8) | **0.002*** |

*p <0.05,

** p < 0.001.

Significant measures are highlighted.

Abbreviations: PA, Physical Activity, PE, Physical Education. PSQI, Pittsburg Sleep Quality Index. PHQ, Patient Health Questionnaire.

[a] Eating habits- Frequency options-0-7.

[b] Snacking pattern- Healthy snack items (sandwich/sprouts/popcorn), High sugar foods (Chocolates/Cakes/Pastries/Ice cream), High fat foods (Fried Indian snacks, such as samosa/vadapav), High salt foods (Hakka noodles/fried rice/Manchurian), Fast foods (wafers/chips/French fries/Burger/pizza).

[c] Time spent in screen-based activities- Responses, < 30 min/d to > 4 h/d, scored from 0 to 4.

[d] Screen addiction was assessed on a five-point Likert scale (strongly disagree to strongly agree), scored 0 to 4.

## Discussion

Several key findings emerged from this study– 1) Adolescents' daily screen usage was substantially high with the mean reported ST being higher during weekdays as compared to the weekends. A considerable amount of time was spent using screens for studying/doing homework and a ubiquitous screen addiction behavior was observed in adolescents. 2) Only a few were involved in moderate to vigorous PA levels with the engagement in PA being even lower in girls, highlighting the magnitude of physical inactivity during the pandemic. 3) Skipping breakfast, infrequent family meals, and frequent consumption of fast foods, fried foods, and carbonated beverages was reported. Girls reported a higher prevalence of sleep latency, daytime dysfunction, and poor sleep quality, and the PHQ-2 scores (for risk of depression) were higher among older adolescents, ages 13–15 years. 4) Additionally, associations were observed between screen usage and eating habits, PA, and sleep quality among the sampled adolescents. A higher ST was associated with lower PA and increased sleep problems and a greater screen addiction was inversely related to healthy eating habits, PA, and sleep variables, though not with depression symptoms. At the time of the data collection, the adolescents had been confined to homes for almost a year due to the country-wide lockdown restrictions imposed since March 2020. This is likely to have resulted in substantial disruptions in the daily routines, lifestyle behaviors, and mental wellbeing of adolescents.

In line with our findings, several studies had reported excessive usage of digital devices among adolescents during the pandemic [3,8,18]. The adverse health consequences of excess ST on the risk of obesity, anxiety, depression, and cardiovascular problems are established

**Table 4. Association of screen time during COVID-19 pandemic with eating habits, physical activity levels, sleep quality, and depression symptoms of 10–15 years old adolescents in Mumbai, India.**

| Measures | Eating habits | | Physical activity levels | | Sleep quality | | Depression symptoms | |
|---|---|---|---|---|---|---|---|---|
| | β (SE) | P value | β (SE) | P value | β (SE) | P value | β (SE) | P value |
| Frequency of screen usage/week | | | | | | | | |
| • Laptop/desktop | -0.298 (0.089) | 0.034* | -0.845 (0.212) | <0.001** | 0.612 (0.334) | 0.044* | 0.245 (0.197) | 0.071 |
| • Mobile/smart phone | -0.123 (0.015) | 0.520 | -0.412 (0.330) | 0.112 | 0.546 (0.221) | 0.012* | 0.212 (0.065) | 0.156 |
| • Television | -0.345 (0.135) | 0.024* | -0.118 (0.067) | 0.540 | 0.219 (0.110) | 0.738 | 0.113 (0.101) | 0.718 |
| Time spent in screen activities/day | | | | | | | | |
| • Studying/doing homework | - 0.198 (0.065) | 0.331 | -0.089 (0.056) | 0.566 | 0.329 (0.115) | 0.038* | 0.192 (0.031) | 0.089 |
| • Using social networking sites | - 0.229 (0.110) | 0.044* | -0.312 (0.216) | <0.001** | 0.312 (0.210) | 0.119 | 0.412 (0.220) | 0.018* |
| • Playing games | - 0.116 (0.078) | 0.665 | -0.032 (0.011) | 0.328 | 0.448 (0.178) | 0.002* | 0.210 (0.114) | 0.178 |
| Total screen time/day | | | | | | | | |
| • Weekdays | -0.199 (0.123) | 0.114 | -0.343 (0.130) | 0.006* | 0.547 (0.312) | 0.046* | 0.125 (0.063) | 0.348 |
| • Weekends | -0.224 (0.112) | 0.082 | -0.831 (0.119) | 0.002* | 0.198 (0.148) | 0.321 | 0.099 (0.045) | 0.648 |
| Screen addiction behaviors | | | | | | | | |
| • I can't imagine going anywhere without my mobile devices | -0.121 (0.110) | 0.412 | -0.512 (0.215) | 0.043* | 0.298 (0.089) | 0.005* | 0.166 (0.103) | 0.118 |
| • Screen time takes up majority of my leisure time | -0.658(0.222) | 0.017 | -0.220 (0.119) | 0.128 | 0.210 (0.116) | 0.421 | 0.101 (0.049) | 0.332 |
| • I prefer socializing online than meeting people face to face | - 0.088 (0.034) | 0.882 | -0.240 (0.112) | 0.019* | 0.254 (0.188) | 0.026* | 0.034 (0.012) | 0.614 |
| Model Summary | Adj r² = 0.133, F = 1.909, Significance = 0.012 | | Adj r² = 0.154, F = 4.728, Significance = 0.002 | | Adj r² = 0.142, F = 2.973 Significance = 0.026 | | Adj r² = 0.080, F = 1.410, Significance = 0.136 | |

Abbreviations: β, standardized regression coefficient. SE, standard error.

*Significant at p <0.05.

**Significant at p <0.001.

Eating habits indicate mean scores of eating habits (breakfast, family meals, eating out, ordering takeaways) related items and snacking pattern (frequency of healthy and unhealthy snack consumption) related items.

Physical activity levels indicate mean summary PAQ-C/A (Physical Activity Questionnaire for Children and Adolescents) scores.

Sleep quality indicates mean global PSQI (Pittsburg Sleep Quality Index) scores.

Depression symptoms indicate mean PHQ 2 (Patient Health Questionnaire-2) scores.

All models were adjusted for sex, age, type of school attended (private vs government schools).

[6,7,22,31]. However, there is also a growing interest to explore the use of screens as coping measures for learning, connecting with people, curbing boredom, and having better access to scientific information. A recent study observed that adolescents who were more active in social media were better equipped to handle pandemic induced environmental stressors [32], a report published in the child and adolescent health section of the Lancet journal recommended that the amount of time spent using screens should be tailored keeping into consideration other factors, such as snacking behaviors and activity patterns in adolescents [33] and a systematic review noted that the ST can be leveraged to extend better social and emotional support to children [34]. Given the established negative effects of increased ST and a need to make the most of the time spent using screens for educational and social benefits, it is imperative that age-appropriate resources that encourage judicious use of screens are emphasized and public awareness regarding the adverse health effects of excess ST are simultaneously promoted.

In this study, we observed that the extended screen usage coincided with other unhealthy lifestyle behaviors, such as lower than recommended MVPA, poor eating habits, and inadequate sleep duration and quality in adolescents. The staggering prevalence of physical inactivity observed in our study can be explained by the lockdown that led to restricted access to

organized sports activities, limited free play in playgrounds, parks, and areas around apartments, and a general feeling of fear among parents to send children outside for playing. Adolescents reported increased consumption of unhealthy foods, such as fast foods, fried foods, and carbonated beverages, and also of healthy foods, such as fruits and salads during the pandemic. Emotional overeating and unhealthy snacking behaviors to reduce boredom and stress have been reported in previous studies [35–37]. The finding related to the perceived increase in healthy food consumption concur with similar studies that observed an improvement in the overall diet quality of adolescents due to greater parental involvement in cooking meals at homes and better monitoring of food intake at mealtimes. Regarding sleep and depression variables, more than half of the adolescents had poor sleep quality and 8.6% were observed to be at risk of depression. Alterations in the sleep-wake cycles due to delay in school start time [38], increased vulnerability to anxiety and depression symptoms due to COVID-related fear and social isolation [19], and limited peer interactions to handle academic frustrations and loneliness [39] seem to have further worsened the already pervasive sleep problems and mental health crisis in adolescents globally and India. Similar to our findings, other studies had shown the prevalence of sleep and depression to be higher among females and older adolescents [38,40], indicating that this group might require particular attention during the pandemic.

The associations between screen-related variables and eating habits, PA, sleep, and depression symptoms were explored. Regression analyses indicated that a higher frequency of screen usage and time spent in different screen-based activities and a greater screen addiction behavior score were associated with lower eating habit and MVPA scores and higher sleep disturbances in adolescents. Research has shown a direct relationship between television viewing and unhealthy eating behaviors [6,11,12], increased screen time to displace the time that can be used for engaging in PA [4,17,41], and excess usage of digital devices to suppress melatonin secretion, delay sleepiness, and increase sleep disturbances [17,42,43]. No significant associations were observed between screen-related variables and PHQ 2 scores. These results are inconsistent with previous studies that reported associations between ST and increased mental health problems [8,19], but concur with others that did not observe statistically significant associations between ST, anxiety, and other mental health indicators in youth [44,45]. As the end of the COVID-19 pandemic remains uncertain, the use of digital devices and screens has become an unavoidable necessity. In this context, the role of parents in limiting the sedentary screen-based recreational activities, setting ground rules for screen media usage, and encouraging adolescents to participate in fun indoor activities, such as dancing, rope skipping, playing with hoops, and online fitness classes is of key importance. The families must also use the enforced stay-at-home advisories as opportunities to have more frequent family meals together, model healthier diet and sleep habits, and build better bonds with adolescents to overcome the hardships encountered during the pandemic. Finally, close attention and committed efforts are required from policymakers in India to revisit the ST guidelines, address the emerging challenges of longer screen exposure, inadequate PA levels, and insufficient sleep during the pandemic, and provide feasible solutions to mitigate the associated short and long-term health problems in adolescents.

Despite a fairly large sample size and selection of sample across age and socioeconomic categories (the adolescents, ages 10–15 years were recruited from both private and public schools in Mumbai), our study has a few limitations that must be considered while interpreting the results. Most importantly, the study sites were selected using the convenience sampling method and the study design was cross-sectional, both of which may have limited the generalizability of the findings and temporal associations between screen time and lifestyle behaviors in adolescents. Due to the ongoing pandemic-induced closure of educational institutes in India since late March 2020, conducting an in-person survey was not feasible. So, we had to

conduct an online survey to collect data, which may have introduced a selection bias. Also, the variables were self-reported by the adolescents so can be subject to overestimation/underestimation. Investigating the lifestyle behaviors and risk of depression between adolescents who use ST predominantly for leisure vs school work may help design targeted strategies and evaluating these behaviors across diverse settings (rural and urban), geographical regions, and age categories can guide the development of appropriate health-promoting policies and designing of culturally relevant interventions for improving the physical and mental wellbeing of adolescents in India as elsewhere.

## Conclusions

The present study brought forth a high prevalence of excessive ST, physical inactivity, and poor sleep quality in 10–15 years old Indian adolescents during the ongoing COVID-19 pandemic. Moreover, the results revealed that a greater time spent using screens was associated with a higher engagement in unhealthy eating behaviors, lower PA levels, and increased sleep disturbances in adolescents. It is difficult to predict if these transient behaviors will continue to persist post-pandemic. Nevertheless, amid continued national school closures, the extended screen time and its impact on adolescents' lifestyle behaviors need to be managed effectively. Using screens to involve adolescents in active play, harnessing mobile applications and digital platforms to provide nutrition, PA, and mental health counseling, and involving social media to raise public awareness about adverse health consequences of excess ST present expedient opportunities to support the educational and entertainment needs of adolescents whilst ensuring optimum physical, sleep, and mental health during the pandemic. Proactive parental support for fostering a tighter control on the screen usage, being role models for appropriate ST and PA engagement, ensuring stringent monitoring of adolescents' diet and sleep routines, and prioritizing an early identification of anxiety and depression symptoms in adolescents may add further leverage.

## Acknowledgments

The authors would like to acknowledge Ms. Apurva Agashe for her assistance with the statistical analysis of the data and the heads and/supervisors of schools for providing permission to conduct the study for their students. The authors are thankful to the research team that worked diligently to ensure data quality and the participants of the study for their valuable inputs and cooperation.

## Author Contributions

**Conceptualization:** Panchali Moitra, Jagmeet Madan.

**Data curation:** Jagmeet Madan.

**Formal analysis:** Panchali Moitra.

**Investigation:** Panchali Moitra.

**Methodology:** Panchali Moitra, Jagmeet Madan.

**Resources:** Panchali Moitra.

**Supervision:** Panchali Moitra, Jagmeet Madan.

**Validation:** Jagmeet Madan.

**Writing – original draft:** Panchali Moitra.

**Writing – review & editing:** Panchali Moitra, Jagmeet Madan.

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
