## [Decision Letter · Decision Letter 0]

20 Sep 2021

PONE-D-21-24611Impact of screen time during COVID 19 on eating habits, physical activity, sleep, and depression symptoms: A cross sectional study in Indian adolescentsPLOS ONE

Dear Dr. Moitra,

Thank you for submitting your manuscript to PLOS ONE. After careful consideration, we feel that it has merit but does not fully meet PLOS ONE’s publication criteria as it currently stands. Therefore, we invite you to submit a revised version of the manuscript that addresses the points raised during the review process.

According to the reviewers' comments, particularly the Reviewer #1, I recommend the authors to revise their paper extensively. The invitation to review does not guarantee final publication of the paper.

We look forward to receiving your revised manuscript.

Kind regards,

Kyoung-Sae Na, M.D., Ph.D.

Academic Editor

PLOS ONE

Journal Requirements:

2. Please include additional information regarding the survey used in the study, particularly the sample size calculation, and ensure that you have provided sufficient details that others could replicate the analyses.

Reviewers' comments:

Reviewer's Responses to Questions

**Comments to the Author**

1. Is the manuscript technically sound, and do the data support the conclusions?

Reviewer #1: Partly

Reviewer #2: Yes

2. Has the statistical analysis been performed appropriately and rigorously? 

Reviewer #1: No

Reviewer #2: I Don't Know

3. Have the authors made all data underlying the findings in their manuscript fully available?

Reviewer #1: No

Reviewer #2: Yes

4. Is the manuscript presented in an intelligible fashion and written in standard English?

Reviewer #1: No

Reviewer #2: Yes

5. Review Comments to the Author

Reviewer #1: Summary:

The authors have conducted a well-designed and interesting study, in which the aim to determine the increase in screen-based activities, and thereby screen time, and whether this causes an decrease in physical activity, sleep quality and mental health. The find that the majority of participants report an increase in screen time, as well as unhealthy behaviors. Also, screen time appears to be associated with these unhealthy behaviors. I have however several reservations about the analysis performed, and the structure of the paper. I think the authors should therefore address these reservations to make it ready for publication.

General recommendations:

- If you use related (such as COVID-related or screen-related) there should be a dash (-) between related and the word before related. So, not COVID related or screen related, but COVID-related and screen-related.

- I should consider re-analyzing the risk factors/associations to be able to draw conclusions about the predictive value of your studied variables. As I explain in my comments below, the regression analyses should involve potentially confounding factors. It is too bold to state that for instance screen time was a significant predictors if you don’t account for any other variables. I would suggest performing univariate regression models to determine which baseline variables that were collected show a high correlation, i.e., a p-value below 0.10, and add those to the regression models too. This will make your results way more clinically relevant.

- In contract to what the authors have claimed in the data availability statement, the doi’s given only lead us to the results of the regression analysis and the used questionnaire. As such, the underlying data is not available to the reader. This is not in line with PLOS One’s Data policy.

- I think the level of English within the manuscript could be improved. Although it is for the most part grammatically correct, the constructions of sentences may be improved. Consider to ask a native English speaker to copy-edit the manuscript to improve readability.

Abstract

- The abstract should be changed according to the author changes made during the revision process.

- I would suggested using a structured abstract, as it is more suitable for this type of research in my opinion. This is however the author’s choice.

Introduction:

I think the introduction is a bit chaotic and thereby hard to read through. First, I think the message given can be described much more condense. Also, it would be helpful if the terminology in the introduction would be more uniform, especially screen-time related terms. I think the authors have tried to use the same terminology as used in the papers they refer to, but it would help the reader is findings are summarized. By summarizing findings, it will be much easier to shorten the introduction.

Methods

- “Measures”; I would suggest placing screen time directly below ‘demographics’, as this is your main outcome. This way, it will also be uniform to the results section.

- “The sleep patterns … Index (PSQI)” (lines 195-197) It this questionnaire also validated in adolescents? I could hypothesize that the sleep patterns in adults aren’t necessarily the same as in adolescents.

- Is the Patient Health Questionnaire-2 validated in adolescents?

- “Of 1512 … years (n= 574)” (Lines 207-210); This is a results and should therefore be moved to the Results section.

Results

- Try to be uniform; sometimes participants are referred to as ‘participants’ and sometimes as ‘adolescents’. In would benefit the readability to only use one term.

- “Screen time”: It is not clear to me what is the difference between screen usage and screen time, is there a difference of is it the same. If there is a difference, it should be more clearly described in the methods section, if there is no difference, use the same terms throughout the manuscript.

- I think the generalizability of the results would benefit from adding 95% CI to the percentages given.

- I would suggest the ‘Eating habits and snacking patterns’, ‘Physical activity levels’ and ‘Sleep quality and depression symptoms’ sections to be combined.

- “Figure 1”: I would recommend to place ‘remained same’ before ‘decreased (it is a more logical order that way). Additionally, would recommend to replace ‘Remained same’ with ‘Remained Similar’.

- “Table 3”: I would recommend describing the stratifications used in this table in the ‘Statistical analyis’ section.

- I would suggest add the descriptive results of the ‘Eating habits and snacking patterns’, ‘Physical activity levels’, and ‘Sleep quality and depression symptoms’ in the supplements, as they are now only displayed within the text.

- “Association of ST on lifestyle behaviors”; I think it is too bold to use the term ‘predictor’ in this section. The term association, as used in the subheading, would be more appropriate. Also, you switch between ‘association’ and ‘predictors’ in this section; these terms are not the same. Because there was not adjusted for any confounding factors (especially confounders as age and gender), no conclusions can be drawn about the predictive value of screen time/screen usage on these outcomes. If you are really interested in finding any possible predictors, the authors should study literature to determine for which variables should be adjusted in the analyses. Another way could be to perform univariate regression analyses with as predictors the baseline variables, one by one, and as dependent variables the ‘Healthy eating habit’, ’Physical activity’, ‘Sleep Quality’ and ‘Depression’ variables. This would certainly benefit the reliability of your results.

- “Table 4”: It would be informative to describe in the legend of the table which variables were used for the ‘Healthy eating habit’, ‘Physical Activity’, ‘Sleep Quality’, and ‘Depression’ variables; readers should be able to understand all tables and figures in the manuscript without actually reading the methods or results section.

Discussion

- I think the first paragraph of the discussion can be omitted. The introduction is the section where you highlight the importance of your study, not the discussion. Now it is just a repetition of what you have already written in the introduction. You may briefly repeat the main goal of the study, but thereafter you should write about your most important findings, not about the relevance of the study.

- I have some reservations about your second paragraph, which should become your first paragraph

- In your second key finding (“Results indicated …. during the pandemic”), you claim that the low proportion of adolescents reporting to be involved in moderate to vigorous PA levels highlight the magnitude of physical activity during the pandemic. My first concern is that you refer to percentages which are not shown in the results section nor in the tables or figures. It is an absolute no-go to present new /unmentioned results first in the discussion section. To make this claim, describe this finding first in the results section. Also, you can claim that the proportion of adolescents involved in MVPA is low, but no conclusions can be drawn about the effect of COVID-19 (to do this, you should have retrospectively assessed their involvement in MVPA before COVID-19, but I cannot find this data in your manuscript.

- In your fourth key finding (“Additionally, significant … depression symptoms”: I would advice the authors to choose the appropriate term for your findings, i.e., association, throughout the manuscript. Here, you use three terms, which are not the same.

- “… and 45.1% mentioned the time spent in on or other screen-based activities to be > 6 hours” (lines 341-42). The 45.1% is the exact sum of the percentages, so this means that there was no overlap, is that correct?

- “Regression analysis … in adolescents” (lines 375-377): I would omit the use of predictor as previously explained.

- “Limitations”; I think the recruitment of only Indian adolescent should be considered as a limitation, as it limits the generalizability to other countries.

- “Future studies … in India.” (lines 401-403) I think the authors can broaden this claim; why would it only be interesting to study this in India rather than in other countries?

Conclusions:

- “In summary … the pandemic. Though the … sleep patterns.” (Lines 405-410) I would soften this statement.

- I would suggest shortening the conclusions, and only describe conclusions that can be drawn from your own results. For instance: “As such, … lifestyle behaviors” is not a finding of your study, and shouldn’t be stated in the conclusions section but in the introduction and/or discussion.

Reviewer #2: Here are my comments and suggestions for this interesting and topical paper:

1) The authors should be mindful of wording with regards to the use of PHQ-2 and screening for depression. A PHQ-2 score of >3 is only an indication of likely depression therefore the statement in the abstract “8.6% had depression (PHQ-2 ≥ 3)” is incorrect. Similarly in the discussion, “8.6% reported the presence of depression” (line 367).

2) In the introduction, the authors state that “Studies have consistently shown that the total time spent using screen devices far exceeds the recommended duration among adolescents(6–8)”. It would be interesting to state what the recommended duration is in order to compare to your findings.

3) The authors results show whether the participants’ ST, unhealthy eating patterns, PA have increased/decreased, but do the authors have any quantitative results for this? It would be interesting to quantify these changes comparing pre and during COVID-19 pandemic data if this is available.

4) It would be interesting to note if there was a difference in lifestyle behaviours or depressive symptoms between those adolescents who use ST for predominantly leisure vs predominantly educational purposes?

5) Re-consider the use of the word “dismal” in line 329.

6) Have the authors considered a selection bias in their sample as the surveys were done exclusively online that this would be skewed towards participants that would have higher ST usage?

6. PLOS authors have the option to publish the peer review history of their article (what does this mean?). If published, this will include your full peer review and any attached files.

Reviewer #1: **Yes: **J.H. Vlake

Reviewer #2: No

---

## [Author Response · Author response to Decision Letter 0]

4 Nov 2021

Response to comments related to the journal requirements:

Response: We have formatted the manuscript in line with PLOS ONE’s style requirements.

2. Please include additional information regarding the survey used in the study, particularly the sample size calculation, and ensure that you have provided sufficient details that others could replicate the analyses.

Response: The questionnaire used to collect the data and the dataset analyzed during the current study are available in the [Figshare] [10.6084/m9.figshare.16934107]. We have added the details of sample size estimation between lines 128-132.

RESPONSE TO REVIEWER 1

Reviewer #1: Summary:

The authors have conducted a well-designed and interesting study, in which the aim to determine the increase in screen-based activities, and thereby screen time, and whether this causes an decrease in physical activity, sleep quality and mental health. The find that the majority of participants report an increase in screen time, as well as unhealthy behaviors. Also, screen time appears to be associated with these unhealthy behaviors. I have however several reservations about the analysis performed, and the structure of the paper. I think the authors should therefore address these reservations to make it ready for publication.

Response: Thanks for acknowledging the relevance of the study and for providing the opportunity to further strengthen the manuscript. The suggestions have been noted and a point-by-point clarification on each of the specified aspects is provided.

General recommendations:

1. If you use related (such as COVID-related or screen-related) there should be a dash (-) between related and the word before related. So, not COVID related or screen related, but COVID-related and screen-related.

Response: Noted and revised.

2. I should consider re-analyzing the risk factors/associations to be able to draw conclusions about the predictive value of your studied variables. As I explain in my comments below, the regression analyses should involve potentially confounding factors. It is too bold to state that for instance screen time was a significant predictors if you don’t account for any other variables. I would suggest performing univariate regression models to determine which baseline variables that were collected show a high correlation, i.e., a p-value below 0.10, and add those to the regression models too. This will make your results way more clinically relevant.

Response: Thanks for your valuable feedback. In this study, the associations between screen time and lifestyle behaviors were examined using multivariable linear regression models that were built by applying the backward elimination method (threshold of 0.05 to stay in the model) for selection of predictive variables while controlling for adolescents’ age, sex, and type of school attended (used as a proxy for socioeconomic status) as covariates. Further to your suggestion and our internal discussion with our statistician, we have re-analyzed our data. We have first performed univariable regression analyses to determine the unadjusted effect of factors associated with each of the dependent variables and have then entered the independent variables with a significance level <0.1 into the mixed-effects multivariable regression models to determine the ST variables that were associated significantly with eating habits, PA, sleep and depression symptoms. As indicated in previous studies, we have used the lowest values of Akaike’s and Schwarz’s Bayesian information criteria measures to test the goodness of fit of the final models. Minimal data were missing as we had analyzed data from participants who had provided complete information in the survey. The revised method of statistical analyses is included between lines 211-217 and the revised results are reported in Table 4 as standardized regression coefficients and standard error of coefficients, considered significant at p<0.05 

3. In contract to what the authors have claimed in the data availability statement, the doi’s given only lead us to the results of the regression analysis and the used questionnaire. As such, the underlying data is not available to the reader. This is not in line with PLOS One’s Data policy.

Response: Datasets analyzed during the current study are available in the [Figshare Repository] [10.6084/m9.figshare.16934107]

4. I think the level of English within the manuscript could be improved. Although it is for the most part grammatically correct, the constructions of sentences may be improved. Consider to ask a native English speaker to copy-edit the manuscript to improve readability.

Response: We have revised the text of the manuscript to improve readability.

5. The abstract should be changed according to the author changes made during the revision process. I would suggested using a structured abstract, as it is more suitable for this type of research in my opinion. This is however the author’s choice.

Response: As suggested, the abstract has been provided in a structured format. 

6. I think the introduction is a bit chaotic and thereby hard to read through. First, I think the message given can be described much more condense. Also, it would be helpful if the terminology in the introduction would be more uniform, especially screen-time related terms. I think the authors have tried to use the same terminology as used in the papers they refer to, but it would help the reader is findings are summarized. By summarizing findings, it will be much easier to shorten the introduction.

Response: Thanks for the feedback. We have shortened the introduction to include summarized findings of previous literature and ensured uniformity in the usage of screen time-related terms. 

7. Measures- I would suggest placing screen time directly below ‘demographics’, as this is your main outcome. This way, it will also be uniform to the results section.

Response: Yes, we agree that placing screen time before eating habits and physical activity measures will improve uniformity. Revisions have been made (lines 141-160). 

8. “The sleep patterns … Index (PSQI)” (lines 195-197) It this questionnaire also validated in adolescents? I could hypothesize that the sleep patterns in adults aren’t necessarily the same as in adolescents.

Response: Thanks for the question. Yes, several studies have established the validity and reliability of PSQI to identify sleep problems in adolescents. We have included an additional sentence in the methods to clarify the same (Line 190-191). 

9. Is the Patient Health Questionnaire-2 validated in adolescents?

Response: The validity of PHQ 2 has been evaluated in adolescents in previous studies. The instrument is a widely accepted screening tool for major depressive disorders among adolescents and youth. This information has been added between lines 195-197.

10. Of 1512 … years (n= 574)” (Lines 207-210); This is a results and should therefore be moved to the Results section.

Response: Noted and revised (Lines 226-229.

11. Results- Try to be uniform; sometimes participants are referred to as ‘participants’ and sometimes as ‘adolescents’. In would benefit the readability to only use one term.

Response: Thanks for your suggestion. We have replaced participants with adolescents in the manuscript.

12. “Screen time”: It is not clear to me what is the difference between screen usage and screen time, is there a difference of is it the same. If there is a difference, it should be more clearly described in the methods section, if there is no difference, use the same terms throughout the manuscript.

Response: In this study, we had used the term ‘screen time’ to indicate the time spent working/ studying/playing using any screen device and the term ‘screen usage’ to indicate different screen devices (such as laptops, mobile phones, television) that were used by the adolescents. We have added this explanation in the methods section (between lines 143-145).

13. I would suggest the ‘Eating habits and snacking patterns’, ‘Physical activity levels’ and ‘Sleep quality and depression symptoms’ sections to be combined.

Response: Thanks for the suggestion. We have combined the results of eating habits, physical activity levels, sleep, and depression-related variables under a single subheading (Lines 246-258)

14. “Figure 1”: I would recommend to place ‘remained same’ before ‘decreased (it is a more logical order that way). Additionally, would recommend to replace ‘Remained same’ with ‘Remained Similar’.

Response: Noted and revised Fig 1.

15. “Table 3”: I would recommend describing the stratifications used in this table in the ‘Statistical analysis’ section.

Response: Between lines 203-205, we have mentioned that comparison of variables was performed stratified by sex and age categories (10-12 years and 13-15 years)

17. I would suggest add the descriptive results of the ‘Eating habits and snacking patterns’, ‘Physical activity levels’, and ‘Sleep quality and depression symptoms’ in the supplements, as they are now only displayed within the text.

Response: The descriptive results of eating habits, snacking patterns, physical activity levels, sleep quality and depression symptoms of adolescents, stratified by sex and age categories are provided in Table 3.

18. “Association of ST on lifestyle behaviors”; I think it is too bold to use the term ‘predictor’ in this section. The term association, as used in the subheading, would be more appropriate. Also, you switch between ‘association’ and ‘predictors’ in this section; these terms are not the same. Because there was not adjusted for any confounding factors (especially confounders as age and gender), no conclusions can be drawn about the predictive value of screen time/screen usage on these outcomes. If you are really interested in finding any possible predictors, the authors should study literature to determine for which variables should be adjusted in the analyses. Another way could be to perform univariate regression analyses with as predictors the baseline variables, one by one, and as dependent variables the ‘Healthy eating habit’, ’Physical activity’, ‘Sleep Quality’ and ‘Depression’ variables. This would certainly benefit the reliability of your results.

Response: We agree that the term association is better suited to indicate the relationship between the explanatory ST variables and the dependent variables such as healthy eating habits, physical activity, sleep quality, and depression symptoms as compared to the term predictor. Hence, we have replaced the term ‘predictor’ with ‘association’ in the results. As mentioned previously, the data was re-analyzed and the revised results of mixed effect multivariable regression models adjusted for sex, age, and type of school attended are reported in revised Table 4.

19. “Table 4”: It would be informative to describe in the legend of the table which variables were used for the ‘Healthy eating habit’, ‘Physical Activity’, ‘Sleep Quality’, and ‘Depression’ variables; readers should be able to understand all tables and figures in the manuscript without actually reading the methods or results section.

Response: Thanks for the feedback. We have added the details of the variables used in the regression analyses as footnotes in Table 4.

20. Discussion- I think the first paragraph of the discussion can be omitted. The introduction is the section where you highlight the importance of your study, not the discussion. Now it is just a repetition of what you have already written in the introduction. You may briefly repeat the main goal of the study, but thereafter you should write about your most important findings, not about the relevance of the study.

Response: The discussion text has been revised to restate the research questions and includes discussions regarding the key findings of the current study.

21. I have some reservations about your second paragraph, which should become your first paragraph

- In your second key finding (“Results indicated …. during the pandemic”), you claim that the low proportion of adolescents reporting to be involved in moderate to vigorous PA levels highlight the magnitude of physical activity during the pandemic. My first concern is that you refer to percentages which are not shown in the results section nor in the tables or figures. It is an absolute no-go to present new /unmentioned results first in the discussion section. To make this claim, describe this finding first in the results section..

Response: In Table 3, row 6 under physical activity levels, we have reported that 12% of adolescents had PAQ score> 2 and 8.9% of girls had PAQ score>2 (p <0.001). As this information was already provided under the Results section (Table 3), we have referred to the same percentages as key findings in the second paragraph of the Discussion to highlight the magnitude of physical inactivity among the sampled adolescents (line 329-331). We have included an additional sentence in text between lines 257-258 regarding physical activity levels of adolescents. The claim that the proportion of adolescents who were engaged in MVPA was low during the pandemic is justified in view of the findings, as mentioned in text and Table 3. 

22. Also, you can claim that the proportion of adolescents involved in MVPA is low, but no conclusions can be drawn about the effect of COVID-19 (to do this, you should have retrospectively assessed their involvement in MVPA before COVID-19, but I cannot find this data in your manuscript

Response: We agree that the impact of COVID 19 on screen time and/or lifestyle behaviors cannot be ascertained due to the cross-sectional design of the study that did not have access to retrospective data to infer the changes brought upon adolescents’ lifestyles by the pandemic. This study aimed to evaluate the impact of screen time during the COVID 19 pandemic (and not the effect of the pandemic on screen time or lifestyle habits) on eating habits, physical activity levels, sleep quality, and depression symptoms in adolescents in India. To avoid potential misinterpretation, we have refrained from the usage of terms such as lower, higher, greater with reference to practices during the pandemic as compared to before COVID 19 in the revised manuscript. We have also explicitly stated the research questions in the introduction of the revised manuscript (Lines 104-106).

23. In your fourth key finding (“Additionally, significant … depression symptoms”: I would advice the authors to choose the appropriate term for your findings, i.e., association, throughout the manuscript. Here, you use three terms, which are not the same.

Response: Revised.

24. “… and 45.1% mentioned the time spent in on or other screen-based activities to be > 6 hours” (lines 341-42). The 45.1% is the exact sum of the percentages, so this means that there was no overlap, is that correct?

Response: The percentage (45.1%) was calculated by aggregating the responses received from 586 adolescents who reported the time spent in one or other screen-based activities to be> 6 hours/d. However, an overlap might have existed between adolescents who spent >6 h/d in different screen-based activities. So, we have revised the sentence to include 33.4% who reported spending >6h/d for studying/ doing homework. 

25. “Regression analysis … in adolescents” (lines 375-377): I would omit the use of predictor as previously explained.

Response: We have replaced the term predictor with associated.

26. “Limitations”; I think the recruitment of only Indian adolescent should be considered as a limitation, as it limits the generalizability to other countries.

Response: This study attempted to address the limited knowledge regarding the potential influences of excess ST on lifestyle behaviors in adolescents in India during the COVID 19 pandemic. Moreover, a comprehensive investigation of eating habits, PA levels, sleep quality, and depressive symptoms during the pandemic among adolescents in India was performed. Given the differences in the lockdown restrictions imposed between countries during the pandemic and that the socio-cultural determinants of lifestyle behaviors tend to vary between adolescents residing in different geographical regions, we believe that the topical nature of this study is the novelty and in fact a strength of the study. 

27. “Future studies … in India.” (lines 401-403) I think the authors can broaden this claim; why would it only be interesting to study this in India rather than in other countries?

Response: Though this study is the first large scale survey to assess the association of screen time during the COVID 19 pandemic with several lifestyle behaviors in Indian adolescents, the results are limited by the convenience sampling method and the urban setting of the survey (the sites were located in the city of Mumbai and may not reflect the impact of ST during pandemic on the lifestyle behaviors of adolescents in rural and smaller cities of the country). Hence, we suggested that future studies must be conducted in diverse settings (rural, semi-urban, and urban areas) and across a broader age category of adolescents in India to better understand the influences of excess ST on lifestyles. This information is particularly pertinent to guide the development of appropriate health-promoting policies and designing age and culturally relevant interventions aimed at improving the physical and mental wellbeing of adolescents. As suggested, we have included this sentence in Discussion (Lines 413-416).

28. Conclusions: “In summary … the pandemic. Though the … sleep patterns.” (Lines 405-410) I would soften this statement. I would suggest shortening the conclusions, and only describe conclusions that can be drawn from your own results. For instance: “As such, … lifestyle behaviors” is not a finding of your study, and shouldn’t be stated in the conclusions section but in the introduction and/or discussion.

Response: Thanks for the suggestion. We have made the conclusion more concise and highlighted the recommendations that can be drawn from the results of this study.

RESPONSE TO REVIEWER 2

Reviewer #2: Here are my comments and suggestions for this interesting and topical paper:

Response: Thanks for your valuable feedback and suggestions to improve the manuscript. We are grateful for your acknowledgment of the relevance of this topical study.

1) The authors should be mindful of wording with regards to the use of PHQ-2 and screening for depression. A PHQ-2 score of >3 is only an indication of likely depression therefore the statement in the abstract “8.6% had depression (PHQ-2 ≥ 3)” is incorrect. Similarly in the discussion, “8.6% reported the presence of depression” (line 367).

Response: We have revised the sentences to indicate that the adolescents with PHQ 2 scores above the established threshold score of 3 were identified to be at risk of depression.

2) In the introduction, the authors state that “Studies have consistently shown that the total time spent using screen devices far exceeds the recommended duration among adolescents (6–8)”. It would be interesting to state what the recommended duration is in order to compare to your findings.

Response: Thanks for the suggestion. We have mentioned that the screen time far exceeds the suggested duration of two hours a day (line 59)

3) The authors results show whether the participants’ ST, unhealthy eating patterns, PA have increased/decreased, but do the authors have any quantitative results for this? It would be interesting to quantify these changes comparing pre and during COVID-19 pandemic data if this is available.

Response: To determine the changes in the frequency and duration of screen usage, and the frequency of intake of specific food items and engagement in different physical activities during the COVID 19 pandemic as compared to before pandemic, the adolescents were asked to report whether they perceived the changes as increased, decreased or remained similar. The responses to these questions generated quantitative data that helped estimate the impact of the pandemic on the selected measures. In Figure 1, we have provided the results of the changes in the frequency of screen usage, eating habits, snacking patterns, and physical activity levels during the pandemic as reported by the adolescents.

4) It would be interesting to note if there was a difference in lifestyle behaviours or depressive symptoms between those adolescents who use ST for predominantly leisure vs predominantly educational purposes?

Response: Indeed, a comparison of the lifestyle behaviors and depression symptoms between adolescents who use screens for leisure vs for studying/doing homework is an interesting line of investigation that will add to the knowledge. However, the questionnaire used in the present study does not allow the stratification in discrete groups of adolescents who engaged in a particular screen-based activity. For instance, the adolescents who reported spending >6 hours/day doing homework may also be using screens for leisure > 6/day. Though the data that we currently have cannot establish these differences, we have suggested this aspect as an area that merits further research (Lines 411-413).

5) Re-consider the use of the word “dismal” in line 329.

Response: Revised.

6) Have the authors considered a selection bias in their sample as the surveys were done exclusively online that this would be skewed towards participants that would have higher ST usage?

Response: Due to the ongoing pandemic-induced closure of educational institutes in India since late March 2020, conducting an in-person survey was not feasible. So, we had to conduct an online survey to collect data, which may have introduced a selection bias. We have added this as a limitation of the study in the discussion section.

---

## [Decision Letter · Decision Letter 1]

22 Dec 2021

PONE-D-21-24611R1Impact of screen time during COVID 19 on eating habits, physical activity, sleep, and depression symptoms: A cross-sectional study in Indian adolescentsPLOS ONE

Dear Dr. Moitra,

Thank you for submitting your manuscript to PLOS ONE. After careful consideration, we feel that it has merit but does not fully meet PLOS ONE’s publication criteria as it currently stands. Therefore, we invite you to submit a revised version of the manuscript that addresses the points raised during the review process.

We look forward to receiving your revised manuscript.

Kind regards,

Kyoung-Sae Na, M.D.

Academic Editor

PLOS ONE

Reviewers' comments:

Reviewer's Responses to Questions

**Comments to the Author**

1. If the authors have adequately addressed your comments raised in a previous round of review and you feel that this manuscript is now acceptable for publication, you may indicate that here to bypass the “Comments to the Author” section, enter your conflict of interest statement in the “Confidential to Editor” section, and submit your "Accept" recommendation.

Reviewer #1: All comments have been addressed

2. Is the manuscript technically sound, and do the data support the conclusions?

Reviewer #1: Yes

3. Has the statistical analysis been performed appropriately and rigorously? 

Reviewer #1: Yes

4. Have the authors made all data underlying the findings in their manuscript fully available?

Reviewer #1: Yes

5. Is the manuscript presented in an intelligible fashion and written in standard English?

Reviewer #1: Yes

6. Review Comments to the Author

Reviewer #1: Review round 2 – Screen time

Overall comments:

Overall, I am very pleased with the effort of the authors during the revisions, and I believe the manuscript has increased tremendously compared to the previous version. I am happy to read that the authors have successfully addressed all my comments.

However, there are a few minor points that have to be addressed prior to this study being ready for publication.

General points of interest

1) I think the authors should carefully check the manuscript for their punctuation. To name some examples: “COVID 19” should be “COVID-19” (Abstract, line 26, line 48; Introduction, line 52…), “some countries, including India had” should be “some countries, including India, had” and there should be a comma prior to each “such as” (for instance line 60 in the introduction).

2) the authors should copy-edit the full manuscript for some minor grammatical errors or missing words (for instance “the transmission of coronavirus” should be “the transmission of the coronavirus” (Introduction, line 52-53)

2) I would suggest to place all reference at the end of a sentence instead of after the part of the sentence the reference relates to. This will improve readability, and the reader can still find the appropriate reference if needed.

Abstract:

I’m glad that the authors have re-written the abstract in a structured way. I think it suits the research better.

I would place the characteristics of the included adolescents in the results section of the abstract (“n=1298, Mage 13.2 (1.1), 53.3% boys”).

Introduction:

I am happy to see that the authors improved their introduction quite a lot. I however still think the authors could be much more concise as the current introduction still counts 935 words. I would like to challenge the authors to limit the word count of the introduction to at most 600 words (preferably less than 500). The introduction should only focus on letting the reader understand why you did this research, i.e., why is this research important.

Also, I would re-arrange the last paragraph of the introduction. I would start with the last sentence (“To the best of our knowledge..”, then the aims and research question (personally, I would not state the research question that specific, but embed it in a normal sentence, such as “We sought to evaluate what the impact of screen time during the COVID-19 pandemic would be on eating habits…”, and thereafter state your specific objectives.

Methods:

I would rephrase the sentence “due to the ongoing … collect data” (lines 118-120) to “An online survey was conducted to collect data, as an in-person survey was not feasible due to the onoing ..”.

When explaining the questionnaire concerning screen time. I would suggest omitting the description of specific items, and add the questionnaire to the supplement and refer to that supplement. Now you spend quite a lot words describing the questionnaire, when I will be more concise and also more comprehensive when just referring to the supplement with the questionnaire.

I think the references after the statement “The PAQ-C/A have been extensively used to evaluate general physical activity levels of children and adolescents in previous studies” (Line 177-179) are missing.

Results:

“Of 1512 adolescents who provided ….”; I would rephrase this sentence, as the adolescents didn’t provide parental consent, but their parents did. I would suggest “Of 1512 adolescents, of whom parental consent was provided, “

In the first paragraph, sometime you use the exact number of adolescents (for instance for the age categories), and sometimes you use percentages (for instance about their living arrangements). I would suggest use the same for the whole paragraph, and I would suggest giving the percentage.

Lines 243-244: respectively should always be placed in the end of the sentence (…for study or entertainment, respectively).

For the headings “Comparison of variables between sex and age categories” and “Association of ST on lifestyle behaviors”: omit the first sentence, as this is information you’ll typically find in the methods and distract from the actual results. Also, I would omit “of variables” as this does not add anything to the subheading. Also, try not to explain how certain scores were calculated in the results section, this is also information that should be in the methods and not be repeated in the results.

In the methods section you write in the statistical analysis that coefficients are presented as standard coefficient and standard error (SE); I however only see the standardized coefficients in the text, so please add the SE’s as standardized coefficients themselves doesn’t tell us much.

Discussion:

The first paragraph nicely summarizes the main findings of the study. I would however not repeat the exact results (such as the mean ST of percentages), and also not explain here how for instance vigorous PA levels were estimated, as this is already explained in the methods.

Try just to repeat the main findings in words rather than in numbers, as you already did that in the results. (For instance, the first key finding could be rephrased as: “First, we found that adolescents use screens for a considerable amount of time, and used their screens more during the week than during the weekend. Second, moderate to vigorous PA levels were reached by only a few adolescents, and was the lowest in girls.”).

Try not to use the word significantly too often. Differences are statistically significant, otherwise you wouldn’t call it a difference, the same applies to an association; if an association would not be statistically significant, you wouldn’t call it an association.

7. PLOS authors have the option to publish the peer review history of their article (what does this mean?). If published, this will include your full peer review and any attached files.

Reviewer #1: **Yes: **J.H. Vlake

---

## [Author Response · Author response to Decision Letter 1]

24 Jan 2022

Dear Academic Editor and Reviewer,

Thank you for providing us the opportunity to submit a revised version of our manuscript titled, ‘Impact of screen time during COVID 19 on eating habits, physical activity, sleep, and depression symptoms: A cross-sectional study in Indian adolescents’ for consideration at the PLOS ONE journal. We thank reviewer #1 for his constructive feedback and valuable suggestions. We are submitting a point-by-point response to the suggestions provided by the reviewer and the manuscript with revisions highlighted in a colored font. An unmarked revised version of our manuscript is also submitted.

Sincerely, 

Authors

RESPONSE TO REVIEWER 1

Reviewer #1: Review round 2 – Screen time

Overall comments:

Overall, I am very pleased with the effort of the authors during the revisions, and I believe the manuscript has increased tremendously compared to the previous version. I am happy to read that the authors have successfully addressed all my comments. However, there are a few minor points that have to be addressed prior to this study being ready for publication.

Response: Thank you so much for appreciating our efforts and providing us the opportunity to further strengthen the paper. We are sincerely grateful for your time, expertise, and insightful suggestions that continue to help us tremendously to improve the manuscript. Each of the minor points highlighted by you has been addressed and a point-by-point response to your suggested revisions is provided.

General points of interest

1) I think the authors should carefully check the manuscript for their punctuation. To name some examples: “COVID 19” should be “COVID-19” (Abstract, line 26, line 48; Introduction, line 52…), “some countries, including India had” should be “some countries, including India, had” and there should be a comma prior to each “such as” (for instance line 60 in the introduction).

Response: Thanks for the suggestion. We have made the necessary punctuation-related revisions throughout the manuscript.

2) The authors should copy-edit the full manuscript for some minor grammatical errors or missing words (for instance “the transmission of coronavirus” should be “the transmission of the coronavirus” (Introduction, line 52-53)

Response: We have edited the text to minimize grammatical errors and improve readability. 

3) I would suggest to place all reference at the end of a sentence instead of after the part of the sentence the reference relates to. This will improve readability, and the reader can still find the appropriate reference if needed.

Response: Thanks for your suggestion. All in-text citations have been placed in parentheses at the end of the sentences except for a few citations that had to be placed mid-sentence as they reflected the results of multiple studies as a part of a single sentence. For instance, reference 4 was placed mid-sentence, immediately after the finding of the study followed by reference 10 that indicated the findings of another study (Lines 69-70).

4) Abstract:

I’m glad that the authors have re-written the abstract in a structured way. I think it suits the research better.

I would place the characteristics of the included adolescents in the results section of the abstract (“n=1298, Mage 13.2 (1.1), 53.3% boys”).

Response: As suggested, the participant characteristics have been added in the results section of the abstract (Line 39).

5) Introduction:

I am happy to see that the authors improved their introduction quite a lot. I however still think the authors could be much more concise as the current introduction still counts 935 words. I would like to challenge the authors to limit the word count of the introduction to at most 600 words (preferably less than 500). The introduction should only focus on letting the reader understand why you did this research, i.e., why is this research important. Also, I would re-arrange the last paragraph of the introduction. I would start with the last sentence (“To the best of our knowledge.”, then the aims and research question (personally, I would not state the research question that specific, but embed it in a normal sentence, such as “We sought to evaluate what the impact of screen time during the COVID-19 pandemic would be on eating habits…”, and thereafter state your specific objectives.

Response: Thanks for your valuable feedback. As suggested, the introduction has been shortened from 934 to ~ 650 words to present a concise summary of the scope, context, and significance of the study. The last paragraph of the introduction has been rearranged and the research question has been restated as the primary aim of the study (Lines 90-97). 

6) Methods:

I would rephrase the sentence “due to the ongoing … collect data” (lines 118-120) to “An online survey was conducted to collect data, as an in-person survey was not feasible due to the ongoing ..”.

Response: Revised (Lines 102-104).

7) When explaining the questionnaire concerning screen time. I would suggest omitting the description of specific items, and add the questionnaire to the supplement and refer to that supplement. Now you spend quite a lot words describing the questionnaire, when I will be more concise and also more comprehensive when just referring to the supplement with the questionnaire.

Response: As suggested, we have referred to the questionnaire as supplemental material (line 131) and have provided a summary of the questions, response options, and scoring methods of screen time related items in the main text between lines 130-138.

8) I think the references after the statement “The PAQ-C/A have been extensively used to evaluate general physical activity levels of children and adolescents in previous studies” (Line 177-179) are missing.

Response: The references have been added (Line 157, References 24-25).

9) Results:

“Of 1512 adolescents who provided ….”; I would rephrase this sentence, as the adolescents didn’t provide parental consent, but their parents did. I would suggest “Of 1512 adolescents, of whom parental consent was provided, “

Response: Noted and revised.

10) In the first paragraph, sometime you use the exact number of adolescents (for instance for the age categories), and sometimes you use percentages (for instance about their living arrangements). I would suggest use the same for the whole paragraph, and I would suggest giving the percentage.

Response: Revised (Lines 213-214).

11) Lines 243-244: respectively should always be placed in the end of the sentence (…for study or entertainment, respectively).

Response: Revised (Line 222).

12) For the headings “Comparison of variables between sex and age categories” and “Association of ST on lifestyle behaviors”: omit the first sentence, as this is information you’ll typically find in the methods and distract from the actual results. Also, I would omit “of variables” as this does not add anything to the subheading. Also, try not to explain how certain scores were calculated in the results section, this is also information that should be in the methods and not be repeated in the results.

Response: Noted and revised.

13) In the methods section you write in the statistical analysis that coefficients are presented as standard coefficient and standard error (SE); I however only see the standardized coefficients in the text, so please add the SE’s as standardized coefficients themselves doesn’t tell us much.

Response: Thanks for your feedback. The standardized regression coefficients and standard error values as mentioned in Table 4 have been added in the text and highlighted in red font (Lines 293-301)

14) Discussion:

The first paragraph nicely summarizes the main findings of the study. I would however not repeat the exact results (such as the mean ST of percentages), and also not explain here how for instance vigorous PA levels were estimated, as this is already explained in the methods. Try just to repeat the main findings in words rather than in numbers, as you already did that in the results. (For instance, the first key finding could be rephrased as: “First, we found that adolescents use screens for a considerable amount of time, and used their screens more during the week than during the weekend. Second, moderate to vigorous PA levels were reached by only a few adolescents, and was the lowest in girls.”).

Response: As suggested, we have rephrased the key findings to include only the summary of the main results in the first paragraph of the discussion section (Lines 304-309). 

15) Try not to use the word significantly too often. Differences are statistically significant, otherwise you wouldn’t call it a difference, the same applies to an association; if an association would not be statistically significant, you wouldn’t call it an association.

Response: Thanks. The required revisions have been made throughout the paper.

---

## [Decision Letter · Decision Letter 2]

21 Feb 2022

Impact of screen time during COVID-19 on eating habits, physical activity, sleep, and depression symptoms: A cross-sectional study in Indian adolescents

PONE-D-21-24611R2

Dear Dr. Moitra,

We’re pleased to inform you that your manuscript has been judged scientifically suitable for publication and will be formally accepted for publication once it meets all outstanding technical requirements.

Kind regards,

Kyoung-Sae Na, M.D., Ph.D.

Academic Editor

PLOS ONE

Additional Editor Comments (optional):

Reviewers' comments:

Reviewer's Responses to Questions

**Comments to the Author**

1. If the authors have adequately addressed your comments raised in a previous round of review and you feel that this manuscript is now acceptable for publication, you may indicate that here to bypass the “Comments to the Author” section, enter your conflict of interest statement in the “Confidential to Editor” section, and submit your "Accept" recommendation.

Reviewer #1: All comments have been addressed

2. Is the manuscript technically sound, and do the data support the conclusions?

Reviewer #1: Yes

3. Has the statistical analysis been performed appropriately and rigorously? 

Reviewer #1: Yes

4. Have the authors made all data underlying the findings in their manuscript fully available?

Reviewer #1: Yes

5. Is the manuscript presented in an intelligible fashion and written in standard English?

Reviewer #1: Yes

6. Review Comments to the Author

Reviewer #1: I congratulate the authors for doing a good job addressing my comments and revising the manuscript.

Some minor grammatical issues remained, but can be solved in the copy-editing stage.

7. PLOS authors have the option to publish the peer review history of their article (what does this mean?). If published, this will include your full peer review and any attached files.

Reviewer #1: **Yes: **Johan H. Vlake

---

## [Editor Report · Acceptance letter]

28 Feb 2022

PONE-D-21-24611R2 

Impact of screen time during COVID-19 on eating habits, physical activity, sleep, and depression symptoms: A cross-sectional study in Indian adolescents 

Dear Dr. Moitra:

I'm pleased to inform you that your manuscript has been deemed suitable for publication in PLOS ONE. Congratulations! Your manuscript is now with our production department. 

Kind regards, 

on behalf of

Dr. Kyoung-Sae Na 

Academic Editor

PLOS ONE